# Assessment of ^18^F-PBR-111 in the Cuprizone Mouse Model of Multiple Sclerosis

**DOI:** 10.3390/diagnostics11050786

**Published:** 2021-04-27

**Authors:** Valerie L. Jewells, Hong Yuan, Joseph R. Merrill, Jonathan E. Frank, Akhil Patel, Stephanie M. Cohen, Ben Giglio, Nana Nikolaishvili Feinberg, Glenn K. Matsushima, Zibo Li

**Affiliations:** 1Department of Radiology, University of North Carolina at Chapel Hill, Chapel Hill, NC 27599, USA; yuanh@med.unc.edu (H.Y.); zibo_li@med.unc.edu (Z.L.); 2Biomedical Research Imaging Center, University of North Carolina at Chapel Hill, Chapel Hill, NC 27599, USA; Joseph.r.merrill@gmail.com (J.R.M.); Johnathan_Frank@med.unc.edu (J.E.F.); giglio@email.unc.edu (B.G.); 3UNC Neuroscience Center, University of North Carolina at Chapel Hill, Chapel Hill, NC 27599, USA; akhilp@live.unc.edu (A.P.); glenn_matsushima@med.unc.edu (G.K.M.); 4Lineberger Comprehensive Cancer Center, University of North Carolina at Chapel Hill, Chapel Hill, NC 27599, USA; stephanie_iovino@med.unc.edu (S.M.C.); nana_nikolaishvili@med.unc.edu (N.N.F.); 5Department of Microbiology and Immunology, University of North Carolina at Chapel Hill, Chapel Hill, NC 27599, USA; 6Integrative Program Biological and Genome Sciences, University of North Carolina at Chapel Hill, Chapel Hill, NC 27599, USA

**Keywords:** ^18^F-PBR-111, cuprizone, multiple sclerosis, microglia, TSPO

## Abstract

The study aims to assess site assessment of the performance of ^18^F-PBR-111 as a neuroinflammation marker in the cuprizone mouse model of multiple sclerosis (MS). ^18^F-PBR-111 PET imaging has not been well evaluated in multiple sclerosis applications both in preclinical and clinical research. This study will help establish the potential utility of ^18^F-PBR-111 PET in preclinical MS research and future animal and future human applications. ^18^F-PBR-111 PET/CT was conducted at 3.5 weeks (*n* = 7) and 5.0 weeks (*n* = 7) after cuprizone treatment or sham control (*n* = 3) in the mouse model. A subgroup of mice underwent autoradiography with cryosectioned brain tissue. T2 weighted MRI was performed to obtain the brain structural data of each mouse. ^18^F-PBR-111 uptake was assessed in multiple brain regions with PET and autoradiography images. The correlation between autoradiography and immunofluorescence staining of neuroinflammation (F4/80 and CD11b) was measured. Compared to control mice, significant ^18^F-PBR-111 uptake in the corpus callosum (*p* < 0.001), striatum (caudate and internal capsule, *p* < 0.001), and hippocampus (*p* < 0.05) was identified with PET images at both 3.5 weeks and 5.0 weeks, and validated with autoradiography. No significant uptake differences were detected between 3.5 weeks and 5.0 weeks assessing these regions as a whole, although there was a trend of increased uptake at 5.0 weeks compared to 3.5 weeks in the CC. High ^18^F-PBR-111 uptake regions correlated with microglial/macrophage locations by immunofluorescence staining with F4/80 and CD11b antibodies. ^18^F-PBR-111 uptake in anatomic locations correlated with activated microglia at histology in the cuprizone mouse model of MS suggests that ^18^F-PBR-111 has potential for in vivo evaluation of therapy response and potential for use in MS patients and animal studies.

## 1. Introduction

Currently, poor imaging outcomes and assessment measures plague the study of prognostic determinants of multiple sclerosis (MS) [1]. Insensitive MRI measures that only assess anatomic sequela of the disease processes, but not molecular pathophysiology, are a probable issue. Therefore, imaging that assesses cellular activity and molecular information is needed. Microglia/macrophages (MM) provide central nervous system (CNS) immune surveillance influencing inflammatory and reparative processes with short-term neuroprotective activity. These MM are involved in long-term pro-inflammatory cytokine induced neuronal degeneration and short-term inflammatory response to brain injury/BBB disruption, both processes in MS [2,3]. Secondary resultant synaptic stripping triggers subsequent phagocytosis with the release of TNF-alpha, nitric oxide (NO) and glutamate-induced damage [4]. Hence, in vivo imaging of neuroinflammation associated with MM levels would provide a window into patient specific pathophysiology and assist with animal testing. In vivo detection of microglia is now possible with PET ligands of 18-kDa translocator protein (TSPO), also called peripheral benzodiazepine receptor (PBR) [5]. TSPO is a nuclear encoded mitochondrial protein significantly expressed in the MM of MS, the uptake of which correlates with atrophy, a known prognostic indicator in MS [6]. Each non-FDA approved agent requires site-specific validated radiopharmaceutical production and animal assessment prior to human use.

The cuprizone intoxication model (CIM) of MS has been utilized for several decades for noninvasive monitoring of sterile inflammatory CNS demyelination and re-myelination, showing features similar to MS post oligodendrocyte damage [7,8,9,10,11,12]. CIM demonstrates 50% demyelination at 3.5 weeks of cuprizone feeding (CF), which is accompanied by increasing numbers of activated MM [7]. When complete demyelination of the corpus callosum (CC) (a reproducible site of demyelination at the level of the fornix) is achieved at 5 weeks, the MM numbers are maximal in C57BL/6J mouse CIM [8]. These changes occur despite visible enhancement, intact blood brain barriers, or resultant abnormal axonal diameters and internode space post remyelination [9,12]. This is a short-coming of conventional MRI. Therefore, the CIM has great potential for the study of microglial and astrocytic interactions, including microglial phagocytic debris clearance and macrophage recruitment stepping-stones for future MS treatment screening in the preclinical stage [5,6,10,12]. Because TSPO shows high expression in MM, we conducted PET imaging with TSPO ligands in the CIM. The CIM provides an ideal model to track MM and determine the clinical efficacy of TSPO probes that target activated immune cells with the potential to lead to a reliable imaging tool to assess new therapeutic interventions [9,10,13,14,15,16].

The recent development of radioisotope labeled TSPO ligands, such as ^11^C-PK11195 and ^18^F-PBR-111, allows assessment of MM induced inflammation. One TSPO ligand, [^123^I]-CLINDE, has been investigated in the CIM, revealing significant increased uptake during demyelination and decreased uptake with re-myelination, but is limited by less-optimal uptake and biodistribution. Previously, Van Camp et al. demonstrated that ^18^F-PRB-111 displayed better selectivity and affinity binding than ^11^C-PK11195 and ^11^C-CLINME [5,14]. Additionally, the longer half-life of ^18^F compared to ^11^C yields a longer time for image acquisition and greater ease of use and is therefore the targeted TSPO agent for our research. Detailed PBR-111 chemical information can be found in Pubchem (https://pubchem.ncbi.nlm.nih.gov/substance/56431711#section=Top; accessed on 1 July 2009). ^18^F-PBR-111 has been used to assess the experimental autoimmune encephalitis (EAE) model of MS [16,17]. Although not currently FDA approved, ^18^F-PBR-111 has been evaluated in humans and has demonstrated a positive correlation between uptake of ^18^F-PBR111 and MS lesion score [15]. More validation studies are needed to demonstrate its specificity and sensitivity in preclinical models for expanded human applications. Therefore, we propose to characterize ^18^F-PBR-111 imaging performance in the CIM and compare PET imaging to MM immunostaining to provide in-depth information for future application in both preclinical and clinical MS diagnosis and treatment response.

## 2. Materials and Methods

### 2.1. Animal Model

This study was approved by the Institutional Animal Care and Use Committee (IACUC). Six-week-old C57BL/6J male mice (*n* = 17) were purchased from Jackson Laboratory (Bar Harbor, ME) and allowed to acclimate prior to cuprizone exposure at eight weeks of age. At 8 weeks, 14 mice were exposed to 0.2% cuprizone in powder feed (CF) while additional mice (*n* = 3) were provided normal chow for 3.5 or 5 weeks.

### 2.2. Radiopharmaceutical Preparation

^18^F-PBR-111 radiosynthesis, as shown below (Figure 1), was performed according to a prior protocol with slight modification [18]. The tosylate precursor of PBR111 (2.0 mg from Aberjona Laboratories, Woburn, MA, USA) was mixed with a solution of ^18^F-tetrabutylammonium fluoride in 0.2 mL acetonitrile. After being concentrated with nitrogen blow, the mixture was heated at 90 °C for 5 min. The crude product was diluted with 5% acetic acid and passed through an alumina cartridge prior to semi-preparative HPLC purification. The ^18^F-PBR-111 was obtained at 39.5 ± 9.4% radiochemical yield (non-decay corrected) with specific activity of 3.9 ± 0.4 Ci/μmol. The collected fraction was diluted with 10 mL water and trapped on a C18 cartridge (200 mg sorbent). Water (10 mL) was passed through the cartridge to remove residual acetonitrile and trifluoroacetic acid (TFA). The product was eluted from the cartridge with ethanol (1 mL) and collected fraction-wise. The most concentrated fraction was diluted with saline to reach a 10% ethanol concentration. The final product was analyzed by analytical HPLC and the identity confirmed through co-injection with standard. The radiochemical purity was >98%.

### 2.3. PET/CT/MR Imaging and Image Analysis

PET/CT and MR imaging was conducted on CF mice (*n* = 73.5 week CF mice, and *n* = 7 5 week CF mice) and control mice (*n* = 3) with anesthesia using isoflurane (1.5%) mixed with oxygen. T2-weighted MR RARE sequences were performed on a 9.4T MR scanner (BioSpec 94/30 model, Bruker BioSpin Inc., Billerica, MA, USA) (TE/TR = 30/3323 msec, 200 × 200 matrix size, 0.1 × 0.1 mm in-plane resolution and 0.5 mm slice thickness), and PET/CT imaging (eXplore Vista, GE Healthcare Inc., Waukesha, WI, USA) was conducted over subsequent days. Temperature and respiration were monitored (M1025L, Integrated Circuit Systems, Renesas Electronics America Inc, Durham NC, USA, sourced by SA Instrument, Inc Stony Brook, NY, USA). CT scans were acquired immediately before PET scans for attenuation correction and anatomical registration. A single dose of ^18^F-PBR-111 (10.5 ± 0.4 MBq) was injected into each animal via the tail vein. Two mice in each group had dynamic PET imaging for 60 min, and all others had static PET imaging at 40 min post probe injection. PET images were reconstructed using 2D OSEM algorithms with random, scatter, and attenuation corrections. Standardized uptake value (SUV) was calculated voxel-wise based on injection dose and animal body weight.

For analysis, MRI and PET/CT images from the same animal were registered manually using VivoQuant software. Regions of interest (ROIs) were manually drawn around the CC, hippocampus (H), and striatum (caudate/internal capsule, S) using T2-weighted MR images and then overlaid to the corresponding PET images. Mean SUVs were obtained from these ROIs to form time activity curves in dynamic PET data or report uptake levels at 40 min post injection in static PET data.

### 2.4. Post Imaging Tissue Preparation

Mouse brains were collected, snap-frozen in liquid nitrogen post sacrifice, and coronally cryosectioned to obtain CC, S, and H sections for autoradiography (AR), and for IF staining. AR sections were exposed to a storage phosphor screen overnight and scanned using a digital storage phosphor imaging system (Cyclone Plus, PerkinElmer Inc., Shelton, CT, USA). Sections for IF staining were fixed in 4% formalin at room temperature for 10 min and stored in a −80 °C freezer for further staining.

### 2.5. Histology

Demyelination assessment was performed after fixing using Luxol fast blue-periodic acid Schiff-staining (LFB-PAS) on 5 μm thick coronal slices through the anterior, mid, and posterior CC. CC myelinated axons were scored on a scale of 0 to 3. A score of 3 for a fully myelinated (blue axonal fibers) CC down to 0 for complete demyelination (pink axonal fibers). LFB-PAS sections were scored blinded, sections were averaged, and differences among the groups compared to untreated mice.

### 2.6. IHC Staining and Digital Slide Scanning

A rat monoclonal antibody against F4/80 was purchased from eBioscience (San Diego, CA; 14-4801-8), and rabbit polyclonal antibodies against CD11b were purchased from Novus Biologicals (Littleton, CO, USA; NB110-89474). Triplex immunofluorescence (IF) was carried out on the Bond fully automated slide-staining system (Leica Biosystems Inc., Buffalo Grove, IL, USA). Slides were de-paraffinized in Bond dewax solution (AR9222) and hydrated in Bond wash solution (AR9590). Antigen retrieval for F4/80 was performed for 5 min in Bond Enzyme retrieval (AR9551). After pretreatment, slides were first incubated for 30 min with F4/80 antibody (1:100) followed with ImmPRESS anti-rat HRP antibody (Vector labs, Burlingame, CA, USA; MP-7444-15) and TSA Cy3 (PerkinElmer, Boston, MA, USA). Antigen retrieval for CD11b was then performed for 10 min at 100 °C in Bond-epitope retrieval solution 1 pH 6.0 (AR9961). Slides were incubated in anti-CD11b antibody for 30 min (1:1500), which was detected with Dako Envision Rabbit (Carpinteria, CA, USA; K4003) and TSA Cy5. Nuclei were stained with Hoechst 33258 (Invitrogen, Carlsbad, CA, USA). The stained slides were mounted with ProLong^®^ Diamond Antifade (Mountant Life Technologies, Carlsbad, CA, USA; P36961). Slides were scanned with the Aperio FL scanner (Aperio Technologies, Vista, CA, USA) in the DAPI, AF488, Cy3, and Cy5 channels at an apparent magnification of 20×, and digital images were stored in the Aperio Spectrum eSlide Database.

### 2.7. Analysis of Digital IHC Images and Autoradiography Images

Using the Tissue Align module of Visiopharm software (Version 6.7; Hoersholm, Denmark) images of brain sections stained for CD11b and F4/80 were aligned with corresponding autoradiographs. An unsupervised K-means classifier was used to separate AR images into three (treated samples) or two (controls) ROIs. High intensity regions, typically composed of brain ventricles and folded tissue, were not analyzed. Regions corresponding to medium and low intensity signals on AR images were separately analyzed on IF images for the number of cells positive for CD11b and/or F4/80. An Image Analysis module (Visiopharm) was used to detect and enumerate cells positive for CD11b and F4/80, as well as cells that co-expressed both markers. Spatial correlation between AR and CD11b (or F4/80) stained images was also conducted. The immunostaining images were registered and scaled to the corresponding autoradiography resolution (42.3 μm/pixel) with Gaussian smooth (kernel size of 3). Pixel-pixel correlation was conducted using Image J software to measure the spearman ranked correlation coefficients. The spearman ranked correlation is more appropriate than the common Pearson’s correlation because of potential non-linear relationships between the AR and staining images.

### 2.8. Statistical Methods

Prism 5.0 software (Graph Pad) for Windows was used to perform statistical analyses. SUVs of three ROIs (surrounding the CC, H, and S) were compared to the control, 3.5-week, and 5.0-week CF groups using single factor ANOVA analysis. Staining levels of CD11b and F4/80 were compared between positive uptake and negative uptake regions using the non-parametric Mann-Whitney U test; *p* values of <0.05 were assumed to be significant.

## 3. Results

The cuprizone intoxication model consistently shows demyelination in the CC beginning at week 3.5 and ending in full demyelination by week 5.0, most consistently at the level of the fornix. Figure 2A shows control mice not exposed to Cuprizone, revealing fully myelinated (blue stained fibers) within the CC, whereas mice exposed for 5 weeks were fully demyelinated, showing negligible amounts of myelin (Figure 2B). Furthermore, mice not exposed to cuprizone showed no detrimental effect on normal myelinated CC. Importantly, mice exposed to cuprizone for 3.5 week (faint amount of blue-stained fibers) demonstrate partial demyelination (Figure 2D,E), while full demyelination is noted post 5.0 weeks of CF as well as increased cellularity from MM (Figure 2B). These data suggest the induction of demyelination occurred as predicted and suggest the ^18^F-PBR-111 had no obvious detrimental effects on a myelinated CC. The results imply that sterile inflammation manifested by MM accumulation is present in CF mice.

Using ^18^F-PBR-111, 7 CF mice were imaged at 3.5 weeks and 7 CF mice were imaged at 5 weeks post CF treatment, with 3 control mice imaged at each time point. Figure 3A shows the PET/CT images in the control, 3.5 week, and 5.0 week groups, demonstrating increasing ^18^F-PBR-111 uptake in the 3.5 and 5.0 week animals. Dynamic SUV uptake levels of ^18^F-PBR-111 in Figure 3B show initial rapid vascular input followed by tissue binding and washout dynamics. The signal was stabilized around 40 min post injection. Retention of the probe is much higher in the 3.5 week and 5.0 week CF mice compared to the control mice. Statistical comparison between groups was not conducted because only two mice in each group underwent dynamic PET/CT scans. The quantitative measurement and statistic comparison was conducted in static PET acquisition at 40 min post injection. The ROIs (surrounding the CC, S, and H regions) were defined in the co-registered T2-weighted MRI in Figure 4A, and the mean SUVs of the measured ROIs reveal a significantly higher uptake of ^18^F-PBR-111 in 3.5 week and 5.0 week CF mice compared to the uptake in the control group (*p* < 0.05) (Figure 4B). There was a trend of higher uptake in the 5.0 week CF mice compared to that in the 3.5 CF week mice; however, this did not reach a significant level.

AR was conducted after PET imaging in a subgroup of animals (*n* = 3 from each group). Figure 5A depicts AR images from control, 3.5, and 5.0 week CF mice, demonstrating a significantly high uptake of ^18^F-PBR-111 in CF mice compared to the uptake in the control mice. Uptake ratio between the ROIs (surrounding the CC, S, and H) and the cortex (considered reference tissue with a low uptake level) reveals significant differences between the 3.5 (or 5.0 week) CF mice and the control group. Additionally, there were more significant differences between CF and control mice in the CC (*p* = 0.0016) and S (*p* = 0.0048) regions than in the H (*p* = 0.0289) region (Figure 5B).

Figure 6A–D, shows the AR image and the corresponding neighboring slide stained with CD11b and F4/80. Spatial correlation between positive uptake of ^18^F-PBR-111 and positive staining of CD11b and F4/80 is visible. Quantification from the immunostaining analysis is displayed in Figure 7. The cell density of the positive stained CD11b region was (34.7 ± 4.7) % in ^18^F-PBR-111 positive uptake regions vs. (2.97 ± 1.18) % in the ^18^F-PBR-111 negative uptake region for the 3.5 week CF group, and (31.5 ± 6.9) % vs. (0.4 ± 0.1) % for the 5.0 week CF group. The cell density of positive F4/80 staining was less than the positive cell density of the CD11b-staining. There were no statistically significant differences in both CD11b and F4/80 staining between 3.5 weeks and 5.0 weeks, which corroborated with PET and AR images.

The spatial correlation between AR images and CD11b (or F4/80) immunostaining images was conducted in the whole brain sections, as shown in Figure 8. Both CD11b and F4/80 staining had strong positive correlation to the AR images. The Spearman’s correlation coefficient was 0.64 ± 0.15 for the AR-CD11b pairs and 0.56 ± 0.18 for the AR-F4/80 pairs, indicating slightly higher positive correlation with the CD11b compared to the F4/80 staining.

## 4. Discussion

We have demonstrated significant ^18^F-PBR-111uptake, particularly in the CC, S (caudate and putamen), and H in CF mice, with correlative positive staining of CD11b (for activated microglia) and F4/80 (for macrophages) as neuroinflammation markers in the corresponding regions. In general, in the CIM model, the CC at the level of the fornix is considered to be the most affected region with significant demyelination over time, most easily scored. In this study we looked at the entire CC. Additionally, the finding of increased uptake in the S and H region is new information with regard to the extent of pathological changes. We therefore decided to include all these regions due to ease of assessment comparison of imaging and histology. This inclusion of all regions may have however resulted in wash-out or masking of the counts in the peri-fornix CC, which is a small region causing the lack of statistically significant difference between the 3.5 and 5.0 week CF mice. As expected, demyelination in the CF model increased from 50% at about 3 weeks to close to 100% at 5 weeks post CF treatment in the CC, despite no significant difference in ^18^F-PBR111 uptake between the 3.5 and 5.0 week PET imaging time points, which was further confirmed by AR and IF with CD11b and F4/80 staining. Spatial correlation between the AR and IF imaging demonstrated close correlation between ^18^F-PBR-111 uptake and neuroinflammation. Our results suggest that ^18^F-PBR-111 PET imaging can be used as a reliable neuroinflammation marker, correlating well with CD11b and F4/80, but might not be accurate for demyelination status. A larger group for analysis may however yield more significance, as will future evaluation with MR spectroscopy and DTI.

MM serve many roles in response to cytokines and T-cells, which are known components of disease expression in MS [2,3,4]. Correlative histologic findings in our CIM substantiate MM concentration at these sites; the CC in particular. It is known that in the CIM MS model, peripheral macrophages reside within lesions and, compared to a prior study using [^123^I]-CLINDS [14], in the CIM, our AR uptake appears to be more robust. This could be related to an additional week of cuprizone intoxication (6 weeks in their study). Our study also revealed significant uptake at 3.5 weeks in the CC, H, and S. Future comparison of the two markers may be useful. We also demonstrated increased AR uptake of ^18^F-PBR-111 at 3.5 and 5.0 weeks, which correlates with previous work, as inducible and predictable demyelination-remyelination results in calculable uptake [7]. In the CNS, it is known that there are two different phenotypes of microglia M1 (pro-inflammatory) and M2 (anti-inflammatory) serving different roles in MS [19], but there is no definitive method of differentiation between these two populations with markers. In addition, further understanding of the interaction of astrocytes and microglial recruitment via CXCL10 upregulation as well as removal of damaged myelin could be a future project using TSPO agents with the GFAP-thymidine kinase transgenic mouse line [20]. There is also a need to understand the interaction of oligodendrocytes in MS and their impact upon myelin integrity using TSPO agents in conjunction with DTI, while the use of spectroscopy helps to establish the effects of and amount of neuronal loss [21,22]. DTI and spectroscopy was not assessed in this study due to cost constraints. Comparison of these imaging methods will be performed in future studies with additional funding.

After this study with ^18^F-PBR-111 TSPO in the CIM, there is better understanding about the expected behavior in three regions (CC, H, and S) and uptake parameters to be later applied in human imaging with ^18^F-PBR-111. Future imaging in MS patients may yield individualized evaluation of time-specific disease severity and determination of prognosis, as Datta et al. demonstrated with correlation to the Multiple Sclerosis Functional Composite, as well as known conventional MRI parameters such as black holes [22,23]. Recent research has demonstrated significant promise with TSPO agents in progressive MS clinically isolated syndrome in the cortex of MS patients that correlates with disease severity, suggesting TSPO agents have the potential to correlate better with disease severity compared to conventional MRI, making visible hidden disease and assisting in patient individual response analysis to therapy [23,24,25]. In future human studies, correlation of TSPO uptake with DTI to assess demyelination and axonal changes may prove informative with regard to disease progression. TSPO agent utility, however, extends beyond MS because similar pathophysiologic changes are also seen in ALS, Psychiatric disorders, Alzheimer’s, Parkinson’s, and Huntington’s disease, as well as colitis [26,27]. Therefore, our research group and others will be watching the long-term potential of TSPO agents for the study of many human diseases.

Limitations of our study include lack of direct comparison of this TSPO agent with other TSPO agents. We hope to address this in future projects now that this study has successfully characterized imaging features in the CF MS mouse model. Our study is also limited with regard to relatively low ^18^F-PBR-111 contrast in the brain compared to the extra-cranial tissues. We believe this is secondary to the strong partial volume effects (low resolution nature of PET imaging) on the small positive regions surrounded by normal tissue in the mouse model. This will be less of an issue in humans because more white matter volume is present. We also did not see a statistically significant increase of ^18^F-PBR-111 uptake between 3.5 and 5.0 week CF mice both in AR and PET images, despite a trend, consistent with the IF staining (Figure 6). This is different from the previous study reported by Hiremath (one member of that prior research group is an author on this paper) [8]. These differences could be multi-factorial. Inclusion of a larger brain region for measurement and different neuroinflammation markers and the use of RCA-1 (a general macrophage/microglia marker in their paper) as opposed to CD11b and F4/80, which are more sensitive to microglia subpopulations and not as sensitive to macrophage populations, could account for the differences. Another possible limitation to the future utility in human studies of all TSPO markers including ^18^F-PBR-111 are rs6971 polymorphism variability, secondary to age, and genetic variation, secondary to the binding affinity ratios [28]. This must be tested for and considered in the study design [28]. Our institution also recently assessed the ^18^F-PBR-111 ligand in the EAE mouse model and, therefore, we can in the future compare the two models with respect to treatment effect.

With ^18^F-PBR-111, there is the ability to assess pathophysiologic MM changes with imaging in the animal models of MS. Additionally, the determination of prognosis and disease extent in MS patients has been limited because lesion number/load, magnetization transfer, and grey/white matter atrophy are insensitive/nonspecific. Also limiting is the required expensive and time-consuming computer analysis for comparison over time. Therefore, TSPO agent’s ability to measure visible in vivo MM activity without the need for computer assessment appears promising.

## 5. Conclusions

Our study demonstrated significant ^18^F-PBR-111 uptake in the CC, S and H, which correlated with neuroinflammation staining using CD11b and F4/80, in both 3.5 and 5.0-week post CF compared to the control mice. The difference between 3.5 and 5.0 weeks was not significant, but there was a trend of increase. These findings suggest PET ^18^F-PBR-111 imaging has significant potential as an in vivo imaging marker of neuroinflammation in the mouse CIM and potential for human assessment of microglial activity in MS and other diseases.

## Figures and Tables

**Figure 1 diagnostics-11-00786-f001:**
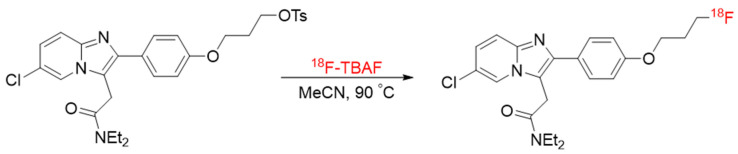
Chemical diagram of radiosynthesis of ^18^F-PBR-111.

**Figure 2 diagnostics-11-00786-f002:**
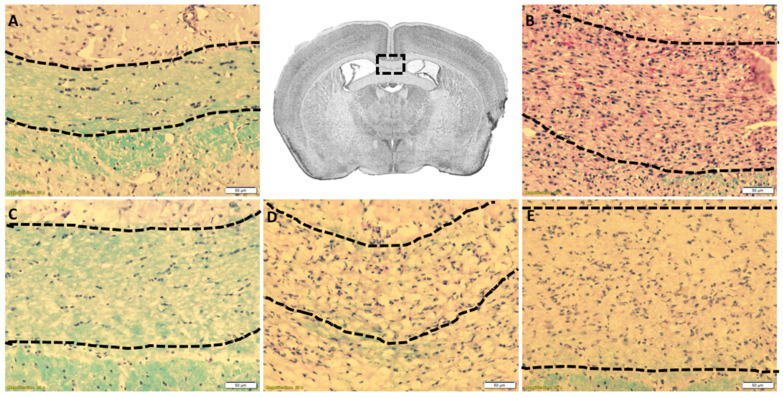
Cuprizone-induced demyelination of the corpus callosum in ^18^F-PBR-111 injected mouse brain sections. Representative image in the first row, center box, demonstrates the area of analysis. LFB-PAS staining of control (**A**) and ^18^F-PBR-111 injected (**B**–**E**) mice after 0 (**A**,**C**), 3.5 (**D**,**E**), and 5 weeks (**B**) after cuprizone treatment was used to analyze myelin levels (blue) and cellularity from MM (dark purple nuclei). Hand-drawn dotted lines represent the area in which scoring for myelin and cellularity was conducted. White scale bar represents 50 microns.

**Figure 3 diagnostics-11-00786-f003:**
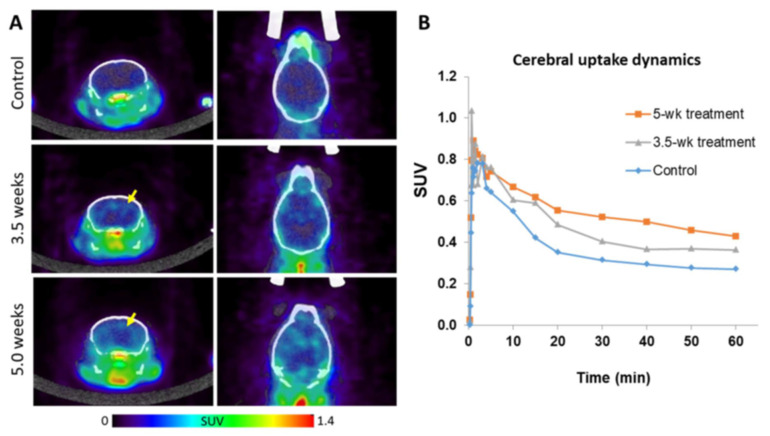
PET/CT images in control, 3.5, and 5.0 weeks cuprizone fed (CF) mice and the uptake dynamics. (**A**) PET/CT images reveal no visible uptake in the control group (top row), but increased brain uptake in 3.5 week (middle row) and 5.0 week (bottom row) CF mice, but less uptake in the brain than in the body (green regions). (**B**) The dynamic uptake curves of ^18^F-PBR-111 in the cerebral region over 60 min post injection demonstrate the initial rapid vascular input followed by tissue binding and washout dynamics. The signal was stabilized around 40 min post injection. Subsequent static PET imaging and quantification was conducted at 40 min post injection.

**Figure 4 diagnostics-11-00786-f004:**
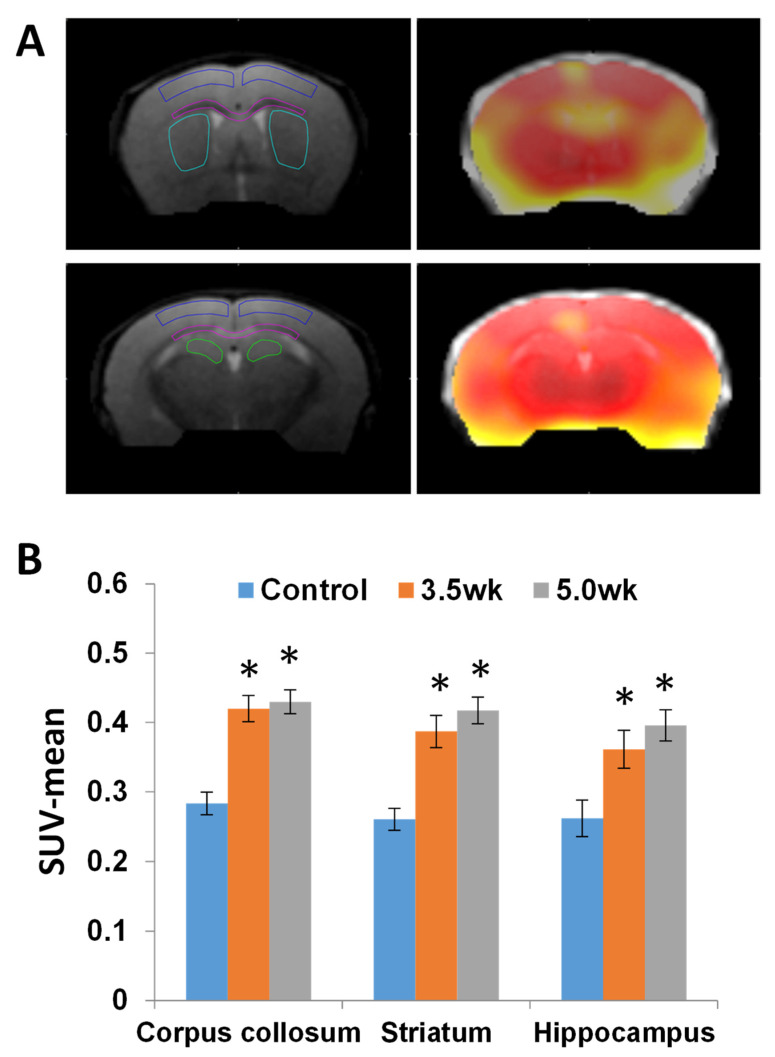
Quantification of the ^18^F-PBR-111 in PET images acquired at 40 min post injection. PET images were registered to the corresponding T2w MR images (**A**). Regions of Interest (ROIs) of cortex (dark blue), corpus callosum (magenta), striatum (caudate/internal capsule, teal/light blue), and hippocampus (green) were drawn in the T2w MRI (left) and superimposed to the PET images (right). There were significant increases (*, *p* < 0.05) of ^18^F-PBR-111 uptake in the corpus callosum, striatum, and hippocampus in 3.5 and 5.0-weeks CF mice compared to the control mice (**B**). No significant differences were detected between the 3.5-weeks and 5.0-weeks groups on the bar graph in the corpus callosum, striatum or hippocampal regions. There was however a trend of increased uptake in all three regions from 3.5 to 5 weeks.

**Figure 5 diagnostics-11-00786-f005:**
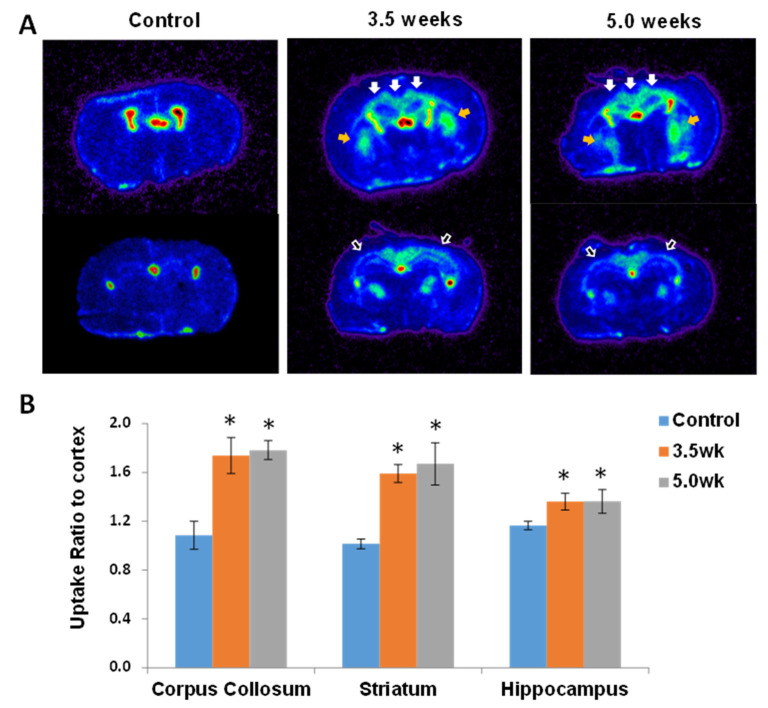
(**A**) ^18^F-PBR-111 autoradiography images of control, 3.5, and 5.0 week cuprizone fed (CF) mice. (**A**) Autoradiography of two brain sections were taken from each mouse to capture the uptake in regions of interest. There was minimal uptake in the control mouse brain, but a strong uptake in the corpus callosum (white arrows) and striatum (orange arrows) at the level of the fornix and the hippocampus lying inferior to the corpus callosum (empty arrows) more posteriorly in 3.5 and 5.0 week CF mice. (**B**) Uptake in these regions were normalized to the cortex, which served as reference tissue. The uptake ratio to cortex for all three regions were significantly higher in the 3.5 and 5.0 week mice compared to controls (*, *p* < 0.05), but no significant difference between the 3.5 and 5 week groups.

**Figure 6 diagnostics-11-00786-f006:**
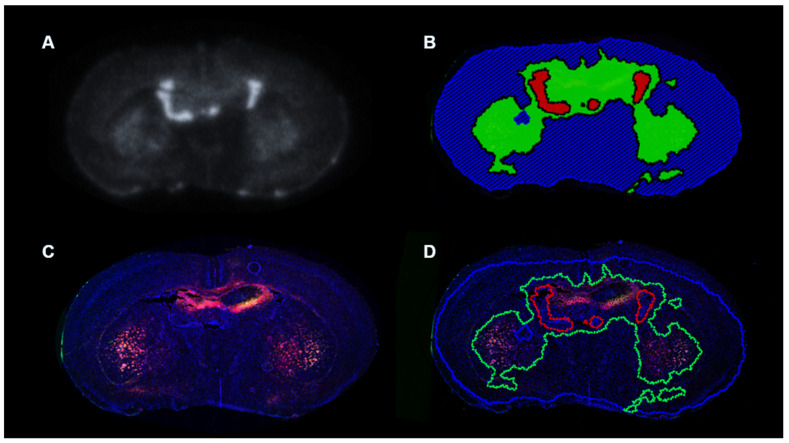
Analysis of autoradiography (AR) and immunofluorescence (IF) images with CD11b and F4/80 staining. Autoradiography image (**A**) was registered to the neighboring images co-stained with CD11b and F4/80 antibodies (**C**) via affine registration using Viziopharm Tissue Align. Images were segmented into low uptake (blue), increased uptake (green), and high uptake ventricle (red, excluded for analysis because uptake is excretion related) regions (**B**), which were utilized to define the ROIs in the IF images (**D**). Positive cell density for CD11b and F4/80 staining was measured in the increased ^18^F-PBR-111 uptake regions (Green, CC, S, and H) and background (surrounding blue).

**Figure 7 diagnostics-11-00786-f007:**
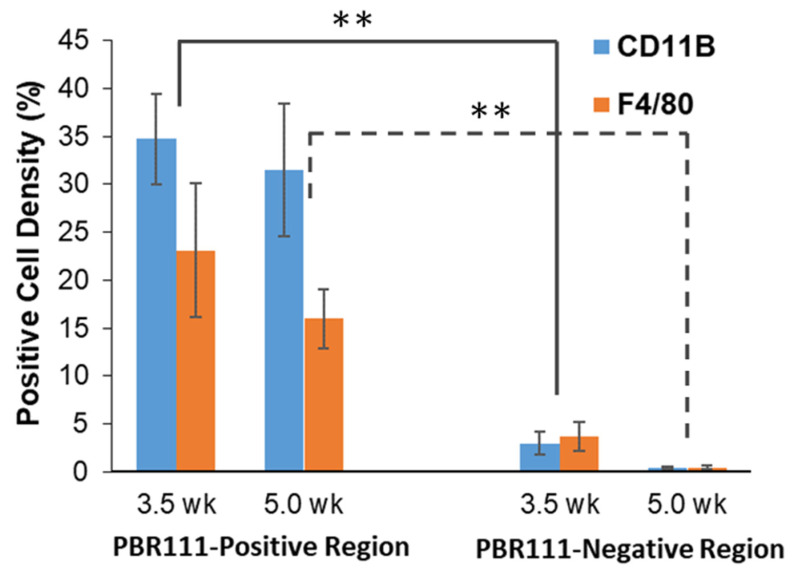
Cell density graph of positively stained CD11b and F4/80 in regions defined as ^18^F-PBR-111 positive region and negative region, as in Figure 5. Regions with high ^18^F-PBR-111 uptake showed significantly higher CD11b positive and F4/80 positive cells (*p* < 0.001). The F4/80 positive cell density was less than the CD11b positive cells in ^18^F-PBR-111 positive region. There was indeed a trend of decreasing microglial and macrophage staining from 3.5 weeks to 5.0 weeks, although there was no significant difference. ** lines defines same week groups, solid for 3.5 week and dotted for 5.0 week groups.

**Figure 8 diagnostics-11-00786-f008:**
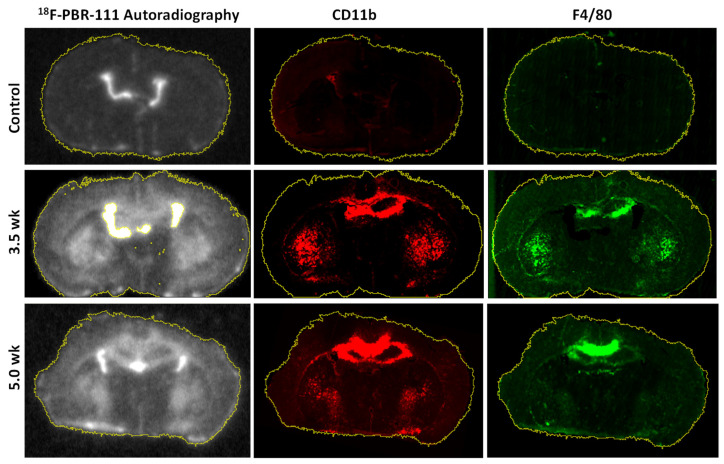
Spatial correlation of registered autoradiography (AR) and immunofluorescence (IF) images of whole brain sections. Background regions and ventricle regions were excluded. In this image, the corpus callosum demonstrates slightly more uptake from 3.5 to 5.0 weeks, while in the caudate and putamen region (S), uptake decreases. The internal capsule and putamen area, however, is increased more at 3.5 weeks, showing that there are more microglia at this location at 3.5 weeks than at 5.0 weeks. Slight differences in C11b and F4/80 may reveal unexplained MM population differences.

## Data Availability

Due to the large volume of image data for PET/CT, MRI and histology as well as data complexity, data is available upon request from the lead author V.L.J.

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
