# Peer review of "Assessment of 18F-PBR-111 in the Cuprizone Mouse Model of Multiple Sclerosis"

_diagnostics, 2021, doi:10.3390/diagnostics11050786_

Round 1

Reviewer 1 Report

Dr. V. L. Jewells et al. report regarding the application of 18F-PBR-111 for detection of TSPO activation in the mouse brain of cuprizone-induced multiple sclerosis animal model. This provides interesting insights, however there are several comments on this manuscript as follows.

  1. In general, it is very difficult for me to consider why the authors conducted such basic and pilot PET study using experimental animals even after the clinical PET study had already reported using same PET probe in multiple sclerosis patients as referred in Reference #15, in which the regions with demyelination were determined using MRI.
  2. In Title, “F18 PRB-111” should be corrected to “18F-PBR-111”.
  3. Throughout the manuscript, “18F-PBR111” should be corrected“18F-PBR-111”.
  4. In Figure-2A, it is very hard for me to identify the brain regions showing significantly higher 18F-PBR-111 uptake in 3.5 and 5.0 weeks cuprizone fed-mice than Control mouse. Considering the time activity curves in Figure-2B, the upper value of color code should be smaller than 1.0 SUV or less.
  5. In Figure-3A, are PET images in right column obtained from Control mouse or 3.5 or 5.0 weeks cuprizone fed-mouse? If it is from Control mouse, it should be nonsense.
  6. In Figure-2B, time activity curves reveal the SUV differences among Control, 3.5 and 5.0 weeks cuprizone fed-mice, however Figure-3B does not indicate any significant differences between 3.5 and 5.0 weeks cuprizone fed-mice. How do authors explain the inconsistency?
  7. In Figure 4A, which regions should be focused for confirmation the results described in the manuscript?
  8. In Results part, authors may be better to evaluate the correlation between PET SUV shown in Figure-3B and Positive cell Density (%) shown in Figure-6 which should be re-analyzed separately among the corpus callosum, hippocampus and striatum.
  9. Are there any evidences to define the relations between demyelination and TSPO activation with accumulated 18F-PBR-111? 9.4T MRI should be applied to confirm the degrees of demyelination by cuprizone administration.

In conclusion, it may be very difficult to conclude that 18F-PBR-111 is better than 18F-FPA-714 as shown in the Reference #18, in which TSPO activation could be detected clearly, and MRI data suggested demyelination and remyelination.

Author Response

Reply to Reviewer #1 Comments

In general, it is very difficult for me to consider why the authors conducted such basic and pilot PET study using experimental animals even after the clinical PET study had already reported using same PET probe in multiple sclerosis patients as referred in Reference #15, in which the regions with demyelination were determined using MRI.  

Response:  In order to ensure appropriate cyclotron radionuclide formation at a specific site of non-FDA approved TSPO agents, each site must validate their utility in an animal study prior to moving to human evaluation. In addition, the use of different agents at different sites is fruitful for determination of whether all TSPO agents will behave in the same manner across sites.  Subsequent comparison at the same site with different agents also requires this initial as you call it basic pilot work to compare and contrast agents in the future. In addition, I am certain that not all readers are aware of the possible utility of TSPO agents in MS and other disease processes since the use of TSPO agents is still experimental at this point in time.

In Title, “F18 PRB-111” should be corrected to “18F-PBR-111”. 

Response: This correction has been made.

Throughout the manuscript, “18F-PBR111” should be corrected“18F-PBR-111”.   

Response: These changes were also made.

In Figure-2A, it is very hard for me to identify the brain regions showing significantly higher 18F-PBR-111 uptake in 3.5 and 5.0 weeks cuprizone fed-mice than Control mouse. Considering the time activity curves in Figure-2B, the upper value of color code should be smaller than 1.0 SUV or less. 

Response:  Yes, this lack of visual difference was not optimal, but is related to the high extra-cranial uptake.  For this reason, autoradiography will be more accurate than PET imaging in animal models. This

This may justify that the 18F-DPA-714 PET agent used by other authors is superior to 18F-PBR-111, but this would require additional funding to perform a comparison of the two agents at the same site using the same mouse strain. Please note that the 18F-DPA-714 PET agent was used with a different mouse stain in the prior referenced study.

In Figure-3A, are PET images in right column obtained from Control mouse or 3.5 or 5.0 weeks cuprizone fed-mouse? If it is from Control mouse, it should be nonsense. 

Response: This is from a 3.5 week CF mouse

In Figure-2B, time activity curves reveal the SUV differences among Control, 3.5 and 5.0 weeks cuprizone fed-mice, however Figure-3B does not indicate any significant differences between 3.5 and 5.0 weeks cuprizone fed-mice. How do authors explain the inconsistency? 

Response:  Yes, there was a trend of higher uptake at 5 weeks compared to 3.5 weeks, but this was not statistically significant using SUV values. Most likely, this is related to the region of ROI that included more than the corpus callosum near the fornix which allowed for ease of comparison to the MRI, PET and histology.  Another possibility includes variable chow consumption leading to different intake of the cuprizone.  In addition, it is also possible that that this agent is less optimal than others for detection of subtle demyelination in mice. This agent however may still be sufficient in human subjects.

In Figure 4A, which regions should be focused for confirmation the results described in the manuscript? 

Response: As previously stated in the legend, the areas of comparison are the corpus callosum, striatum and hippocampi with minimal frontal cortex change. Arrows have been added to depict the regions to improve reader understanding. Please also see edited legend. 

In Results part, authors may be better to evaluate the correlation between PET SUV shown in Figure-3B and Positive cell Density (%) shown in Figure-6 which should be re-analyzed separately among the corpus callosum, hippocampus and striatum. 

Response: An additional image (7) and appropriate legend have been added to respond to this. (Hong Yuan PhD)

Are there any evidences to define the relations between demyelination and TSPO activation with accumulated 18F-PBR-111? 9.4T MRI should be applied to confirm the degrees of demyelination by cuprizone administration. 

Response:  Researchers could attempt to compare TSPO to DTI.  Due to cost constraints for the budget with this project the additional hour needed to perform this on all MRI exams was cost prohibitive. With future studies with more funding, this would be a focus of research particularly, if the goal is to assess response to therapy. This has been added to the discussion.

In conclusion, it may be very difficult to conclude that 18F-PBR-111 is better than 18F-FPA-714 as shown in the Reference #18, in which TSPO activation could be detected clearly, and MRI data suggested demyelination and remyelination.   

Response:  The goal of this project was not comparison of the two agents, but rather establishment of the utility of PBR-111 in our lab for future animal and human investigation as well as affirmation that this process can work in more than one site further supporting the scientific method. In the future, we will attempt to compare more than one agent when funding is available in animal and human subjects.

Reviewer 2 Report

The manuscript by Zibo Li et al. submitted to diagnostics described significant uptake of 18F-PBR-111 in the corpus callosum, striatum, hippocampus which correlated with histologic demyelination and microglial location at histology, suggesting that 18F-PBR-111 exhibits potential for evaluation of therapy response and potential for use in the multiple sclerosis patients. In general, the paper is well written. The experimental methods are described comprehensively and the conclusions are justified by the results. References are adequate. This work is original and is suitable in content for diagnostics. However, it is in need of some minor revisions as suggested below:

(1)   On Page 1, line 12, “A This study” should be “This study”.

(2)   On Page 3, line 91, “n=14 mice” should be “14 mice”.

(3)   On Page 4, line 124, “N=7” should be “n=7”.

(4)   On Page 4, line 125, “N=3” should be “n=3”.

(5)   On Page 4, line 149, “-80C” should be “- 80 °C”.

(6)   On Page 4, line 167, “100°C” should be “100 °C”.

(7)   On Page 9, line 285, “18F-PBR-111” should be “18F-PBR-111”.

(8)   On Page 3, only Radio-HPLC results are afforded, how about are the  results of the radio-TLC?

(9)  In particular, there are several format issues in References section. The manuscript needs proof-reading prior to the final re-submission. 

Author Response

Reply to Reviewer 2

On Page 1, line 12, “A This study” should be “This study”.

Response:  This has been corrected.

On Page 3, line 91, “n=14 mice” should be “14 mice”.

Response:  This has been corrected.

On Page 4, line 124, “N=7” should be “n=7”.

Response:  This has been corrected.

On Page 4, line 125, “N=3” should be “n=3”.

Response:  This has been corrected.

On Page 4, line 149, “-80C” should be “- 80 °C”.

Response:  This has been corrected.

On Page 4, line 167, “100°C” should be “100 °C”.

Response:  This has been corrected.

On Page 9, line 285, “18F-PBR-111” should be “18F-PBR-111”.

Response:  This has been corrected.

On Page 3, only Radio-HPLC results are afforded, how about are the  results of the radio-TLC?

Response:  RadioTLC is not very useful for small molecular weight PET agent. It could not resolve impurity from the product. RadioHPLC provides much better resolution and is a commonly used quality control method to determine the identity and purity of organic PET agents.  (Zibo Li PhD Radiochemist)

In particular, there are several format issues in References section. The manuscript needs proof-reading prior to the final re-submission. 

Response:  These have been corrected.

Round 2

Reviewer 1 Report

As a revised manuscript, the authors sincerely replied to all reviewers’ comments one-by-one.